# Analysis of Consumer Exposure Cases for Alcohol-Based Disinfectant and Hand Sanitizer Use against Coronavirus Disease 2019 (COVID-19)

**DOI:** 10.3390/ijerph19010100

**Published:** 2021-12-23

**Authors:** Hyukmin Kweon, Jae-Won Choi, Seong-Yong Yoon

**Affiliations:** 1Civil and Environmental Engineering, University of California, Los Angeles, CA 90095, USA; kweonhyukmin@gmail.com; 2BenSci Inc., 4822 Elmwood Ave, Los Angeles, CA 90004, USA; 3Environmental Health Center, Soonchunhyang University Hospital, Gumi 39371, Korea; jaewon2125@naver.com; 4Department of Occupational and Environmental Medicine, Soonchunhyang University Hospital, Gumi 39371, Korea

**Keywords:** COVID-19, disinfectant, hand sanitizer, rubbing alcohol, unintended toxic exposures

## Abstract

The development and distribution of vaccines and treatments as well as the use of disinfectants and hand sanitizers to cope with coronavirus disease 2019 (COVID-19) infection has increased rapidly. As the use of disinfectants and hand sanitizers increased, the number of unintended exposures to these substances also increased. A total of 8016 cases of toxic exposure to disinfectants and hand sanitizers were reported to the American Association of Poison Control Centers (AAPCC) from 1 January 2017 to 30 May 2021. The cases have been characterized by substance, sex, patient age, exposure reason and site, treatments received, and outcomes. The number of exposures correlates closely to the rise of COVID-19 cases, rising significantly in March 2020. About half of the total cases involved children less than 10 years old and 97% of those exposures per year were unintentional. In addition, the most common exposure site was the patient’s own residence. Over-exposure to disinfectants and hand sanitizers can cause symptoms such as burning and irritation of the eyes, nose, and throat, coughing, chest tightness, headache, choking, and, in severe cases, death.

## 1. Introduction

Disinfectants and hand sanitizers have served as important components in the U.S. response to the emergence of coronavirus (SARS-CoV-2), which causes coronavirus disease 2019 (COVID-19), both before and after medication or vaccines have been made available [1,2]. The Center for Disease Control and Prevention (CDC) urged the use of alcohol-based hand sanitizer products due to their ability to reduce the transmission of respiratory infection. Similarly, the World Health Organization (WHO) has proposed fighting COVID-19 through preventative measures to maintain safety with healthy lifestyle changes that promote an effective immune system, such as sanitizing hands frequently with alcohol-based hand sanitizers and wearing a mask. Alcohol-based hand sanitizers are mainly made of ethanol, isopropyl alcohols, and hydrogen peroxides in different combinations [3]. Formulations made from these components may become toxic to human health when used in excessive amounts or abused. Based on case reports the Food and Drug Administration (FDA) and CDC received from public health partners in Arizona and New Mexico [4], an FDA consumer alert was issued on 19 June 2020 warning against the use of hand sanitizers that contain methanol (methyl alcohol). During the pandemic time period, the number of people who have experienced the adverse effects of hand sanitizers has nearly doubled [5]. Due to concern about safety for children and adults, the FDA investigated and released a list of hand sanitizer products that should not be used or sold to the general public [6].

The demand for hand sanitizers spiked to prevent the spread of COVID-19. With the CDC recommending alcohol-based hand sanitizers for preventative measures, many suppliers have increased production or even shifted manufacturing lines to produce more alcohol-based hand sanitizers [7]. The growing demand for alcohol-based hand sanitizers is due to advantages which they can provide for reducing the transmission of respiratory infections [8]. Recent research on the risk of excessive use and misuse of such disinfectants and hand sanitizers has been conducted [9]. Hand sanitizers should contain at least 60% ethyl alcohol (ethanol) or 70% isopropyl alcohol in liquid, gel, or foam [10]. Evaporation of these chemicals causes known toxic and hazardous environmental impacts [11,12]. Thus, the recommended unconditional use of disinfectants and hand sanitizers as a precaution against COVID-19 can also potentially be environmentally damaging and pose a significant risk to human health.

The main purpose for this report is to emphasize toxicity and serious health risks to human health from exposure to alcohol-based disinfectants and hand sanitizers. Through identification and characterization of all exposures reported to the American Association of Poison Control Centers (AAPCC) from 1 January 2017 to 30 May 2021, cases were defined by toxicity source, including disinfectant, hand sanitizer, rubbing alcohol, and unknown type alcohol. The point of contact for obtaining AAPCC national data was the regional Washington Poison Center (WPC). In this analysis, we reviewed the cases by substance, sex, patient age, exposure reason and site, treatments received, and outcomes.

## 2. Alcohol-Based Disinfectants and Hand Sanitizers

According to the WHO, an alcohol-based hand sanitizer is “an alcohol-containing preparation (liquid, gel or foam) designed for application to the hands to inactivate microorganisms and/or temporarily suppress their growth. Such preparations may contain one or more types of alcohol, other active ingredients with excipients, and humectants [13].” Additionally, non-alcoholic products are available, but they are less preferred by CDC. This is due to their poor efficacy and narrow spectrum compared with those of alcohol-based sanitization products [14].

Most alcohol-based disinfectants and hand sanitizers contain isopropyl alcohol, ethanol, n-propanol, or a combination of these agents. However, some hand sanitizers containing methanol can cause side effects, such as nausea, vomiting, or headaches. More serious effects include blindness, seizures, or damage to the nervous system if enough methanol is taken internally [15]. Even methanol-free disinfectants and hand sanitizers exposure can cause symptoms such as coughing, chest tightness, burning, and irritation of the eyes, nose, and throat, headache, choking, and, in severe cases, death. Recently, alcohol-based disinfectants used without antibiotic additives proved to be as equally effective as those with additives in reducing hospital-related infections. Active antibiotic additives pose some concern for the possibility of causing dermatitis [16]. Recently introduced alcohol-based foam disinfectants can spread over hand surfaces to have higher compliance and higher efficacy than gel. The efficacy of a hand sanitizer depends on (1) type of alcohol, (2) concentration, (3) quantity applied on hands, and (4) time of exposure [17]. In the case of bacteria sanitization, initial research found more efficacy for less isopropyl alcohol content (50–70% versus 95%) with bacteria number decreased on contaminated hands after hand rubbing [18]. Using more quantitative methods, 65.5% alcohol content was found to be most effective in reducing number of bacteria in the skin [19]. For virus disinfection, 60% alcohol solution is typically recommended, with recent evidence suggesting ethanol and isopropyl alcohol efficiently inactivate the virus SARS-CoV-2 in 30 s at >30% alcohol content. Even 40% alcohol commercial spirits could be suitable as a virus disinfectant if no other options are available in resource limited areas [20].

## 3. The Risk of Ingesting Alcohol-Based Disinfectants and Hand Sanitizers

Despite the proven efficacy of alcohol-based products, the delayed acceptance in some hospitals was due to concerns that repeated use could lead to excessive dryness of the skin [21]. Alcohol kills bacteria and most viruses by removing oil from the skin and instantly modifying its protein structure, but alcohol remains on the skin for at least 30 s [17]. In order to minimize harmful damage from remnant alcohol left on hands, use of disinfectants and hand sanitizers should be combined with regular hand washing. Alcohol containing products not only damage the skin, but dangerous long-term or lethal effects can also occur due to accidental alcohol ingestion prior to its complete evaporation.

Adults, more so than children, can better tolerate small amounts of remnant alcohol from hand sanitizers and disinfectants including methanol, isopropyl alcohol, ethanol, and n-propanol. Methanol concentration of 40–50 mg/dL produces serious toxicity, [15] while an excessive ingested amount of ethanol [22,23] or isopropyl alcohol (125–188 mg/dL) [24] can lead to hospitalization or cause severe internal damage [25]. The body converts methanol to an acid that is toxic to neurons, particularly the retina, and can cause brain injury [26]. Adverse effects from excessive alcohol ingestion can become more severe over time and can be irreversible if left untreated. Effects from alcohol poisoning include drowsiness, reduced level of consciousness (central nervous system depression), confusion, headache, dizziness, inability to coordinate muscle movement, nausea, vomiting, and even heart and respiratory failure. Health and safety concerns for hand sanitizers include (1) accidental ingestion by children and (2) intentional ingestion by adults to become intoxicated. The most adverse outcomes for children concern accidental or intentional hand sanitizer ingestion, which can lead to respiratory arrest that interferes with blood sugar regulation. Dangerous outcomes from inhalation or skin contact with remnant alcohol from hand sanitizers is mostly minimal in adults and children.

## 4. Case Analysis about Consumer Exposure to Alcohol-Based Disinfectants and Hand Sanitizers

### 4.1. Case Collecting Method

We reviewed all exposure case records from 1 January 2017 to 30 May 2021 to identify and characterize disinfectant and hand sanitizer exposure cases in the United States general population. Additionally, rubbing alcohol and unknown type alcohols were investigated. National data requested from the AAPCC and distributed through the regional WPC included reporting of exposures to the poison center on a voluntary basis, not mandated by law. Since AAPCC data describes the number of calls received by the poison center voluntarily, it most likely under-represents the number of actual exposure occurrences. Additionally, exposures do not necessarily represent either poisoning or overdose. Clinical and demographic data extracted from these records are categorized into case definitions by substance, sex, patient age, exposure site, treatments received, and outcomes.

### 4.2. Case Analysis Results and Discussion

A total of 8016 exposure cases to disinfectant, hand sanitizer, rubbing alcohol, and unknown type alcohols are shown in Figure 1. Since 2020, there has been a sudden increase in the number of reports to AAPCC associated with increased use of cleaners and disinfectants indicating the possibility of improper use. Frequent and increased use of disinfectants and hand sanitizers can increase the chances of causing the possibility of accidents as well as health problems. Although the AAPCC data does not provide information showing a clear association between exposure to disinfectants and hand sanitizers due to COVID-19 pandemic and the number of accidents, it appears to have a clear temporary association with increased use of these products. Figure 1 shows cases spiked from 2019 to 2020 with hand sanitizers accounting for the largest number and percentage increase (1080 cases, 176%) compared with disinfectants (376 cases, 152%) and rubbing alcohol (336 cases, 110%).

Figure 2 shows monthly reported cases to AAPCC for exposures to both disinfectants and hand sanitizers. An abrupt 98% increase that occurs at the beginning of March 2020 and temporarily declines in October 2020 has continued to remain high compared with case numbers before the COVID-19 pandemic. In comparison with coronavirus infection case numbers provided by the CDC [27], the temporary decline observed in October 2020 corresponds with the United States government stay at home order during that time.

According to Table 1, males and females had nearly equal case numbers, so there is no correlation between disinfectants and hand sanitizers exposure with gender. When broken down into different age ranges, the increase in total cases was seen across all age groups from 2020 to May 2021. Exposures among children aged less than 10 years represented a significantly high percentage (48%) of total cases between 2017 and May 2021.

Figure 3a shows 97% of exposures were unintentional in children less than 10 years old. Hand sanitizers packaged in colorful bottles or with sweet smells can be tempting to children [28,29]. Misplacement or mishandling of these products, coupled with hand sanitizers having high alcohol content, can lead to poisoning in young children who may become ill from a small amount of alcohol due to their body size. Despite an increase in total cases between 2019 and May 2021, the ratio between intentional, unintentional, adverse reaction, and other exposures remained relatively similar. The age range with the most intentional exposures was between 10 and 19 years old. This trend was prevalent for all years analyzed, with a peak value of 48% intentional exposures in 2020. One explanation is that teenagers may be using alcohol-based disinfectants and hand sanitizers to hurt themselves. In Figure 3b, children under the age of 10 were most exposed to hand sanitizers at 72%, followed by teenagers aged 10 to 19 at 51%. In the case of adults over the age of 20, the cases of disinfectants and rubbing alcohol exposure were relatively higher than those under the age of 19.

Further analysis of where these exposures were happening is summarized in Table 2. The largest percentage increase from 2019 to 2020 among all exposure sites was own residence (87% in 2020), which increased 156% (from 1199 to 3064). This trend continues into 2021; however, data for 2021 was collected only up to May. Exposures at schools have gone down since 2019, most likely due to remote schooling during 2019 and 2020. The number of reported cases of exposure in the workplace increased, which could be due to cleaning workers being exposed by frequent sanitizing. Healthcare facility and restaurant exposures were small in case numbers, but these exposures were mostly due to adverse reactions. Data for treatments on the cases reported are also shown in Table 2 More than 50% of annual reported cases were provided treatment. Unfortunately, from 2017 to May 2021, there is a downward trend of follow up on reported cases. In 2017, 37% of the cases reported had no follow up (393 out of 1058) and that percentage increased to 57% in 2020 (882 out of 1554). The severity of those exposures is also shown in Table 2. More than 50% of the reported cases had minimal or no effect, while the number of major effects was insignificant.

### 4.3. Illustrative Cases

**Case 1.** A 7-month-old boy (7.7 kg) licked about 1 teaspoon of ethanol-based hand sanitizer unintentionally at home. About 10 min later, a report was filed with the AAPCC by his mother. In a health care facility, basic medical treatment was performed and after 4 h of observation, the patient was able to be discharged.

**Case 2.** An 8-year-old boy (31.7 kg) unintentionally drunk an unknown quantity of ethanol-based hand sanitizer at home. The case was reported to AAPCC by his mother in 2 min. Initial laboratory investigations were conducted, and the patient was treated. He recovered after a 3-day hospitalization and was discharged. The medical outcome was not followed up on and further minimal clinical effects were possible.

**Case 3.** A 23-month-old boy (11.3 kg) unintentionally ingested 25 mL of a multipurpose cleaner (unknown type of alcohol) at home. The case was reported to AAPCC by his father in 30 min. Unknown therapy was provided, and the patient was reported to have minor permanent damage.

## 5. Conclusions

Alcohol-based disinfectants and hand sanitizers are one way to clean hands after touching surfaces and/or interacting with people, particularly when handwashing is not an option. In low-resource settings, where soap, water, and alcohol-based sanitizers are not readily accessible, alternative sanitization methods, including sand, soil, ash, soda ash, seawater, alkaline materials, sunlight, and 40% alcohol commercial spirits have also been evaluated. During the COVID-19 pandemic, the CDC and WHO have recommended washing hands frequently with soap and water for at least 20 s, especially after being in public, blowing your nose, coughing, or sneezing. If soap and water are not available, the CDC recommends using an ethyl or isopropyl alcohol-based hand sanitizer. However, excessively frequent, and improper use of disinfectants and hand sanitizers can lead to serious toxicity. This can be due to accidental intake, absorption through skin contact, and suicide attempts. Since 2017, 8016 exposures to disinfectants and hand sanitizers were reported to AAPCC. Between the 3 years of the period from 2017 to 2019, before the pandemic, 3398 cases were reported. Meanwhile in the 1.5 years of 2020 to May 2021 4618 cases were reported. That is over 1000 more cases in half the time. Approximately 50% of these exposures occurred among children aged 10 or less. Among that age group, 96% of these cases were unintentional exposures of disinfectants and hand sanitizers. This may be due to improper location or handling of the disinfection product. From 2017 to May 2021, the age range with the most intentional exposures were for those between the ages of 10–19 years. In this analysis, the most common exposure site was own residence. Over 50% of the reported cases showed minimal to no effect, while the number of major effects was significantly less, at 28 out of 8016. Frequent exposure to disinfectants and hand sanitizers can cause symptoms such as coughing, chest tightness, burning and irritation of the eyes, nose, and throat, headache, choking, and, in severe cases, death. Although most cases resulted in minimal effects, some still required treatment by health care professionals. Public health officials and policymakers should consider all the available evidence to make qualified recommendations, such as child-proofing containers for disinfectants and hand sanitizers. Informing consumers about all the risks associated with these products is essential for adult caregivers to understand how to help keep these substances away from children between times of supervised use.

## Figures and Tables

**Figure 1 ijerph-19-00100-f001:**
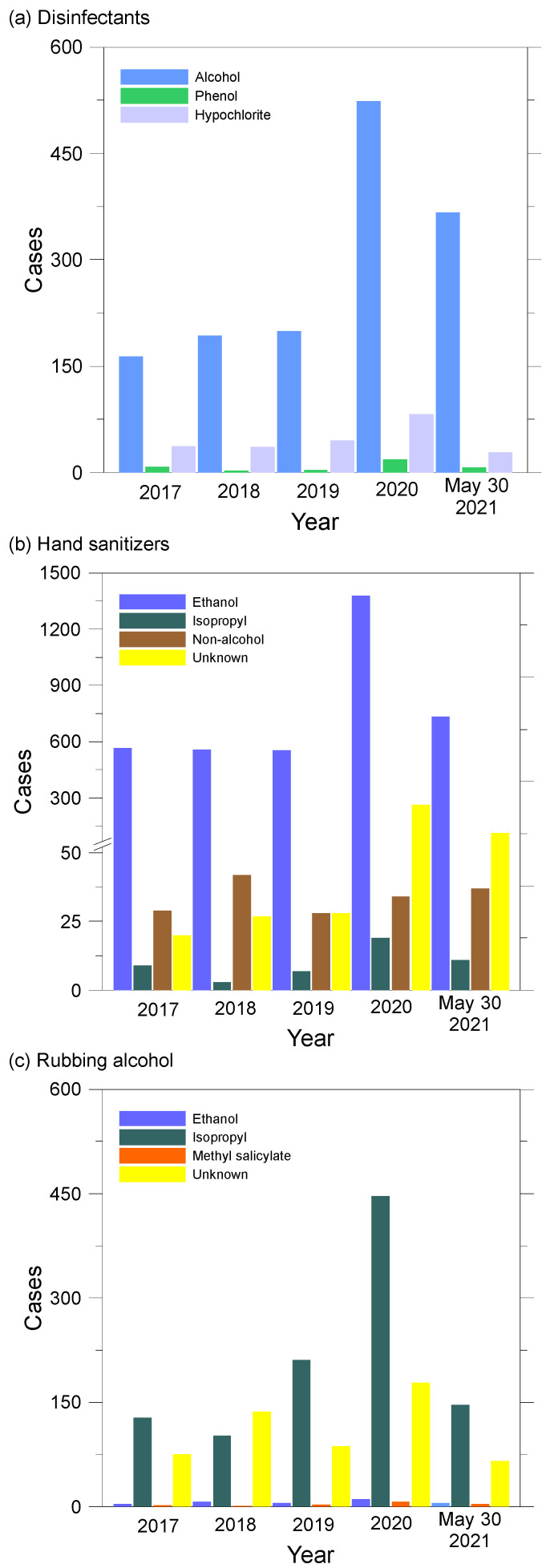
Number of exposures to alcohol-based (**a**) disinfectants, (**b**) hand sanitizers, and (**c**) rubbing alcohol reported to the American Association of Poison Control Centers (AAPCC), arranged by selected characteristics—1 January 2017 to 30 May 2021.

**Figure 2 ijerph-19-00100-f002:**
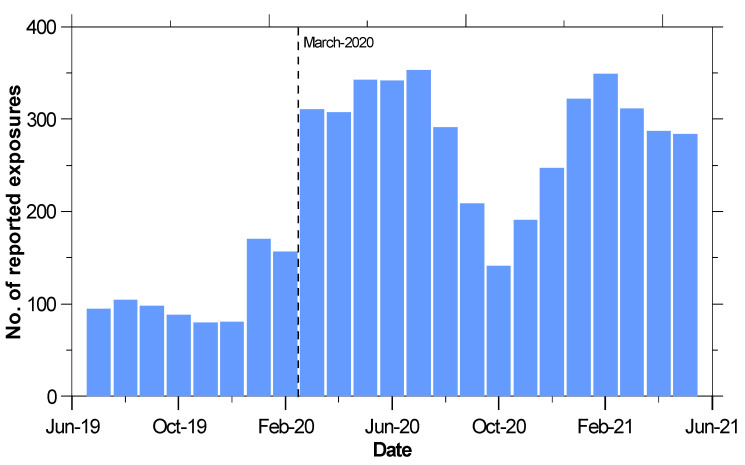
Number of monthly exposures to alcohol-based disinfectants and hand sanitizers reported to AAPCC—United States, July 2019 to May 2021.

**Figure 3 ijerph-19-00100-f003:**
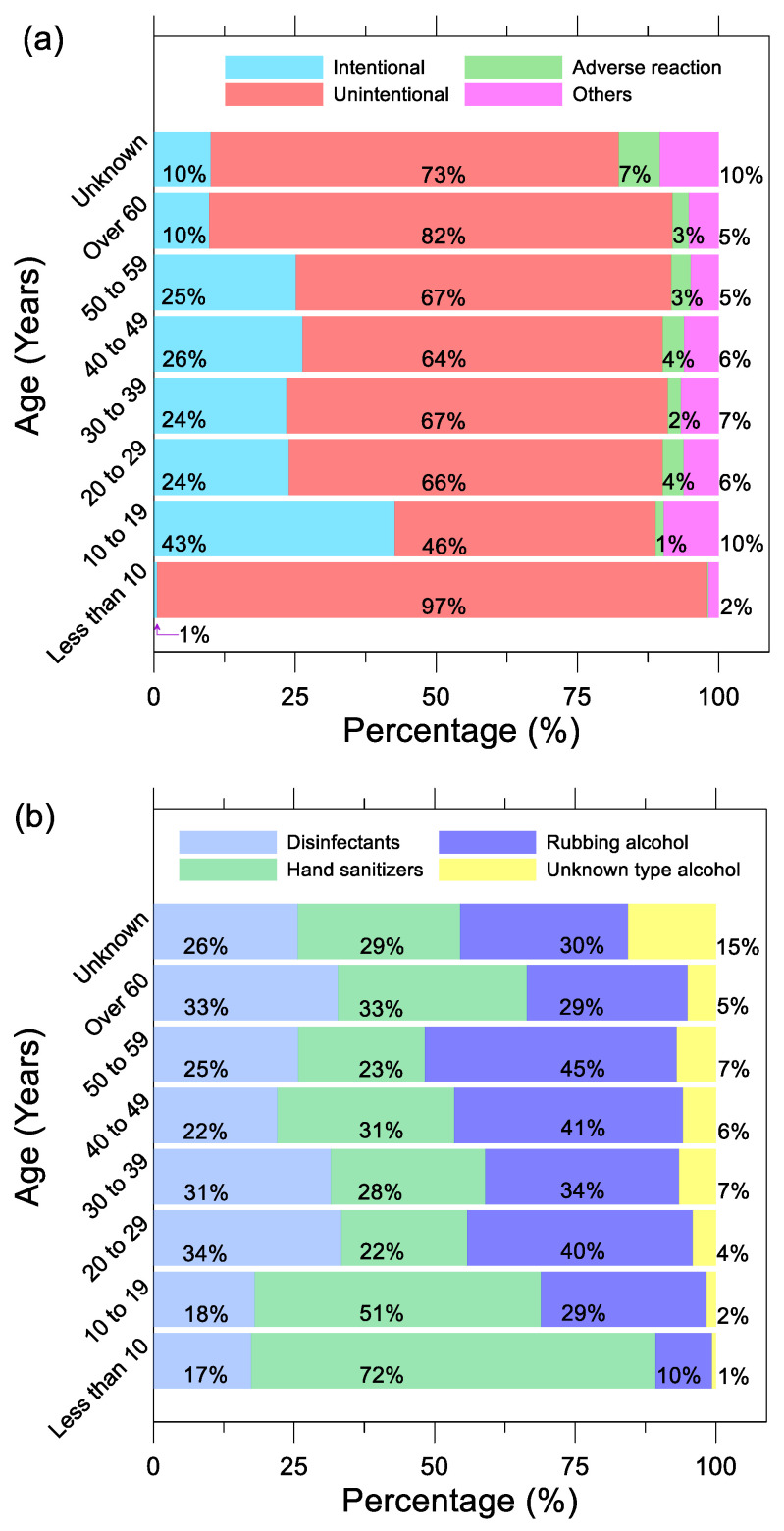
Percentage of different age ranges (**a**) exposure to disinfectants, hand sanitizers, rubbing alcohol, and unknown type alcohols; (**b**) exposure reasons reported to AAPCC from 2017 to May 2021.

**Table 1 ijerph-19-00100-t001:** Yearly number of cases of exposures to disinfectants, hand sanitizers, rubbing alcohol, and unknown type alcohols based on sex and age range.

		2017	2018	2019	2020	May 2021
Sex	Male	524	568	603	1412	714
	Female	531	567	589	1630	834
	Unknown	3	6	7	22	6
	Total	1058	1141	1199	3064	1554
Age (Years)	Less than 10	545	560	565	1312	836
	10 to19	85	102	104	207	127
	20 to 29	94	122	142	323	137
	30 to 39	88	89	107	360	105
	40 to 49	54	67	84	232	75
	50 to 59	51	47	69	193	87
	over 60	71	90	86	292	123
	Unknown	70	64	42	145	64
	Total	1058	1141	1199	3064	1554

**Table 2 ijerph-19-00100-t002:** Number of cases per exposure sites, whether treatment was received, and the outcomes of exposures to disinfectants, hand sanitizers, rubbing alcohol, and unknown type alcohols.

		2017	2018	2019	2020	May 2021
Exposure site	Own residence	903	949	1004	2675	1399
	Workplace	45	59	63	178	57
	Public area	31	27	30	37	32
	School	39	51	41	11	5
	Healthcare facility	4	6	9	10	14
	Restaurants	2	4	3	4	3
	Unknown	34	45	49	149	44
	Total	1058	1141	1199	3064	1554
Treatment received/ Outcome	Therapy provided	601	592	564	1423	624
	Observation only	54	57	62	116	34
	No therapy provided	10	7	6	29	14
	Unknown	393	485	567	1496	882
	Total	1058	1141	1199	3064	1554
	No effect	101	100	105	213	58
	Minimal effect	637	713	722	1631	892
	Minor effect	172	132	141	339	204
	Moderate effect	43	41	44	93	29
	Major effect	2	7	8	11	0
	Unable to follow	103	148	179	777	371
	Total	1058	1141	1199	3064	1554

## Data Availability

All data were extracted on request from the American Association of Poison Control Centers (AAPCC) through the Washington Poison Center. If the original data is required, it can be provided upon request, and the patient’s medical record cannot be provided. Reporting of exposures to the poison center is voluntary and thus not mandated by law. As such, WPC data describes the number of calls received by the poison center and most likely is an under-representation of the true occurrence of any one substance. Exposures do not necessarily represent a poisoning or overdose.

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
