# Peer review of "Analysis of Consumer Exposure Cases for Alcohol-Based Disinfectant and Hand Sanitizer Use against Coronavirus Disease 2019 (COVID-19)"

_ijerph, 2021, doi:10.3390/ijerph19010100_

Round 1

Reviewer 1 Report

The manuscript 'Cases analysis of consumer exposure to alcohol-based disinfectant and hand sanitizer against coronavirus disease 2019' by Kweon et al is of significant importance. The authors have stressed the adverse effects of the over-exposure to hand sanitizers/disinfectants and described the potential sideeffects of using them

The authors should add in the conclusion the alternatives for sanitization apart from the hand washing solution and importance of spreading the awareness of right balance of preventing Covid but also not using the alcohol-based sanitizers excessively.  I recommend the manuscript to be published after minor edits

Author Response

Dear Reviewer,

We would like to thank the reviewer and the editor for the positive and constructive comments and suggestions of our manuscript entitled “Analysis of consumer exposure cases for alcohol-based disinfectant and hand sanitizer use against coronavirus disease 2019 (COVID-19)”. Those comments are very helpful and valuable to revise and improve our manuscript. We have studied the reviewer’s comments carefully and have made revisions which are marked in highlight in the “Revised manuscript”. We have tried our best to revise our manuscript according to the comments and offered specific point-by-point responses to the reviewer’s comments below. We hope that our revision and responses convince you to accept our manuscript for publication.

Regards,

Hyukmin Kweon

Reviewer 2 Report

The authors provide a decriptive evaluation of data on disinfectant exposures from the Washington Poison Center.

The following revisions are necessary:

Title: The title could be more descriptive about the scope, add WPC data

Line 52: It is probably untrue that not much research into this issue is available. For example, the annual evaluation in „Side effects of drugs annual“ has reviewed a large number of studies during the last years: e.g. most recent volume: Lachenmeier DW. Antiseptic drugs and disinfectants with special scrutiny of COVID-19 pandemic related side effects. Side Effects of Drugs Annual. 2021;43:275-284. doi:10.1016/bs.seda.2021.03.001 https://www.ncbi.nlm.nih.gov/pmc/articles/PMC8488688/

Line 73 and throughout: check style for intext references. Should be before fullstop.

Line 87-90: in the context of covid I would have expected some remarks about viruses instead of bacteria. I believe that the SARS-CoV-2 virus is killed by alcohol concentrations as low as 40%. Check: Neufeld, M., Lachenmeier, D. W., Ferreira‐Borges, C., & Rehm, J. (2020). Is Alcohol an “Essential Good” During COVID‐19? Yes, but Only as a Disinfectant!. Alcoholism: Clinical and Experimental Research44(9), 1906-1909.

Section 3, especially lines 97-131: this whole part is completely unscientific and not backed up by references or facts. It should be deleted as it stands now.

Some specific remarks about this section:

Line 99: ethanol due to evaporation does NOT have „dangerous effect“. The inhalatory exposure from evaporated ethanol is almost negligible

Line 116: „all-around just bad“: this is a language not acceptable in a scientific journal

Line 121-131: the claims about children are also not competely plausible. There is some literature that systemic ethanol exposure is only occurring at levels of concern, when skin is lacerated or in pre-term babies.

Section 4: I would restructure the section into „Methods“,“Results“ and „Discussion“ similar to a normal original research paper (also check the journal template)

Line 134: why were only the WPC data used and not the US national data, which also should be available? (I cited some studies about US national data in Lachenmeier (2021), reference above). This must be included and compared in the discussion.

Line 171: how can the accidental exposures in children be explained? Are disinfectants not normally distributed in child-proof containers? Or was there a COVID-regulation that allowed selling them in non child-proof containers?

Line 202 and throughout: I would round off all percentage values tot he full number. The decimals do not make much sense considering the confidence intervals oft he data.

Line 220: all units should be recalculated to metric units (kg)

Line 229: check language, „tasked“?

Line 264: WAPC or WPC?

Line 267: delete Acknowledgements, this is the template text

Author Response

(The authors gave the same response as above.)

Reviewer 3 Report

This is an interesting  evaluation and a  case analysis of consumer exposure to alcohol-based disinfectant and hand sanitizer against coronavirus disease 2019 3 (COVID-19).  The report is well prepared and indicates the abuse of alcohol based sanitizers during the current epidemic. Such data are urgently warranted.

The paper is well written but in some places there are grammatical errors and colloquialism creeping in. Hence a proper English edit is required.

The manuscript can be accepted if the authors can attend to the following.

Title: cases analysis should be case analysis

Line 1:

` Before medication or vaccines were available, disinfectants and hand sanitizers were 29 an important component….. ` This Is wrong as they are still important in mitigating infection spread.

Line 83

The addition of active antibiotics raises concerns of the possibility of dermatitis and unknown long-term effects of residual organisms on skin plants.

What is the meaning of this statement?

Line 99 :….dangerous effect on the inside of the body.

Edit please very colloquial. Suggest: ….. dangerous systemic effects.

Line 140 : Eight thousand sixteen cases of exposure to disinfectant

Suggest: A total of 8016 cases….

Table 1;  Table 1. Number and percentage of exposures to… no %  are  given. It will be good to have  a graphic representation to show the increments of accidents.

Fig 2 is too detailed and  redundant; What are the authors attempting to show here. Could be deleted.

(Figure 2. Percentage of different age ranges’ yearly exposure to disinfectants, hand sanitizers, rubbing alcohol, and unknown type alcohols reported to WPC (a) 2017, (b) 2018, (c) 2019, (d) 2020, and 198

(e) May 2021.)

Conclusions:

The author should shift the material in the Data Availability Statement to the discussion and indicate that the data may be an underestimate as all cases are not reported. Last, but not least, it is good if the authors can suggest some preventive measures to mitigate these accidental poisonings, especially amongst young children.

Author Response

(The authors gave the same response as above.)

Round 2

Reviewer 2 Report

All my comments were adequately considered. Thank you for the revisions and considerable improvements of the paper.